# Risk factors associated with the development of delirium in general ICU patients. A prospective observational study

**Beatriz Lobo-Valbuena**[1,2]*, **Federico Gordo**[1,2], **Ana Abella**[1,2], **Sofía Garcia-Manzanedo**[1], **Maria-Mercedes Garcia-Arias**[1,2], **Inés Torrejón**[1,2], **David Varillas-Delgado**[3], **Rosario Molina**[1,2]

1 Intensive Care Unit, Hospital Universitario del Henares, Coslada, Madrid, Spain, 2 Grupo de Investigación en Patología Crítica, Facultad de Ciencias de la Salud, Universidad Francisco de Vitoria, Pozuelo de Alarcón, Madrid, España, 3 Facultad de Medicina, Universidad Francisco de Vitoria, Pozuelo de Alarcón, Madrid, España

* b.lobo.valbuena@gmail.com

**Data Availability Statement:** The aggregate data underlying this study are available in the manuscript and its Supporting Information files.

## Abstract

### Objective

We aimed to analyze risk factors related to the development of delirium, aiming for early intervention in patients with greater risk.

### Material and methods

Observational study, including prospectively collected patients treated in a single general ICU. These were classified into two groups, according to whether they developed delirium or not (screening performed using CAM-ICU tool). Demographics and clinical data were analyzed. Multivariate logistic regression analyses were performed to quantify existing associations.

### Results

1462 patients were included. 93 developed delirium (incidence: 6.3%). These were older, scored higher on the Clinical Frailty Scale, on the risk scores on admission (SAPS-3 and SOFA), and had a greater number of organ failures (OF). We observed more incidence of delirium in patients who (a) presented more than two OF (20.4%; OR 4.9; CI95%: 2.9–8.2), and (b) were more than 74 years old albeit having <2 OF (8.6%; OR 2.1; CI95%: 1.3–3.5). Patients who developed delirium had longer ICU and hospital length-of-stays and a higher rate of readmission.

### Conclusions

The highest risk observed for developing delirium clustered in patients who presented more than 2 OF and patients over 74 years old. The detection of patients at high risk for developing delirium could imply a change in management and improved quality of care.

There are ethical restrictions on our minimal data set, that were imposed by the Francisco de Vitoria University's Healthcare Ethics Committee (with decision number 44/2018) given that the data contains sensitive identifying information. Dr. Irene Salinas-Gabiña (e-mail address: irenesalinas@hotmail.com) may field data access queries and maintain long-term data accessibility. In order to request our minimal data set, the DOI of the published article should be specified along with the accession code "DeliriumHEN-BLV2021A".

**Funding:** Financial support is provided to cover publication fees, through Fundación para la Investigación e Innovación Biomédica of Hospital Universitario Infanta Sofía & Hospital Universitario del Henares (FIIB HUIS HHEN, Director: Marta Neira Álvarez) in the form of one of the prizes of RESEARCH AWARDS 2020, promoted by this Foundation (certificate issued in Madrid on 12 April 2021), awarded to BLV. No additional external funding was received for this study.

**Competing interests:** The authors have declared that no competing interests exist.

## Introduction

Delirium is a severe neuropsychiatric disorder of organic origin characterized by the appearance of alterations in both consciousness and cognitive functions [1]. The development of delirium is associated with multiple complications: increased mortality [2], longer duration of mechanical ventilation, higher reintubation rate, and increased hospital stay [3–5]. Unfortunately, despite the increasing number of delirium publications in recent years, it remains an underdiagnosed and somewhat underestimated problem [6].

Effective treatment of delirium has proven troublesome. Therefore, prophylactic strategies become paramount. In addition, knowing the different risk factors and the degree of their association with the development of delirium can help identify patients at high risk. In this regard, some of the risk factors identified in previously published studies [7–9] (which are in line with the experience in our center) include: advanced age, personal history of previous high blood pressure or cognitive impairment, urgent surgery, or trauma before admission to the Intensive Care Unit (ICU), high APACHE II score (Acute Physiology and Chronic Health Evaluation) upon admission, and need for mechanical ventilation. Moreover, knowledge of the risk factors associated with delirium development and their implication in the patient's prognosis (both short and long-term) may imply a change in our daily practice [8, 10–13].

The detection of these high-risk patients could reinforce preventive measures. However, it remains to be defined which interventions are the most effective. Clinical guidelines [8] have recommended using a bundle approach (e.g., ABCDEF bundle) to target eliminating multiple modifiable risk factors of ICU, reducing the chances of suffering delirium, or shortening its duration once established. Bundle interventions have been proposed to be more effective than any single-component strategy, but studies are still inconsistent and present contradictory data [14, 15].

Moreover, no pharmacological agent has demonstrated efficacy in treating or preventing delirium. Therefore, the current guidelines [8] suggest against routine use of dexmedetomidine [16, 17], statins [18] or ketamine [19] to prevent delirium. Thus, there is still much to be done and much to be researched to tackle delirium and its consequences. Meanwhile, as healthcare professionals dedicated to critical patient care, we are responsible for identifying and treating the effects of critical illness on patients, both inside and outside the ICU.

Our hypothesis implies the association between the presence of delirium during ICU admission and a worse immediate prognosis. Therefore, our study's primary goal is to analyze the differences between patients who develop delirium and patients who do not develop delirium during ICU admission. Secondarily, we aim to study their characteristics to detect risk factors associated with delirium's appearance during ICU admission and its impact on the patient's early prognosis in our population.

## Materials and methods

We conducted an observational study including prospectively collected data of a cohort of patients admitted to a general ICU from October 1, 2016, up to -and including- May 1, 2019. Our general ICU is in a second-level hospital, including all types of medical patients and 24-hour coverage by general surgeons, urology, orthopedic surgery, and gynecology/obstetrics, excluding cardiothoracic and neurosurgical patients.

Data were collected prospectively in the Registry of the Intensive Care Unit of the University Hospital of Henares. The research was approved by the Francisco de Vitoria University's Healthcare Ethics Committee (44/2018). Participation and acceptance of inclusion of patient's data into the Registry were obtained by signing the informed consent document (by the patient or by an authorized surrogate in case the patient was unable to express their opinion). The study includes all patients admitted during the mentioned period who agreed to participate in

the Registry. Exclusion criteria were patients under 18 years old and patients who required transfer to another hospital (given the impossibility of correct data collection and follow-up upon discharge from the ICU).

We collected relevant demographic and clinical data in every patient, including sex, age, Clinical Frailty Scale (CFS) [20–22], admission type, reason for admission, comorbidities (cardiovascular, respiratory, renal, hepatic, cancer disease, endocrine), specific measures of critical patient management (mechanical ventilation, continuous extrarenal clearance technique, isolation, continuous neuromuscular blockade (NMB), prone) and development of organ failure (s) (OF) during ICU admission. We used the Simplified Acute Physiology Score (SAPS-3) and SOFA score (Sequential Organ Failure Assessment) on admission as validated scores for severity of illness [23, 24]. Organ failure was defined by SOFA score above two as individual scores for each organ, determining progression of organ dysfunction. In addition, invasive mechanical ventilation duration was calculated by adding up the time (in days) of all consecutive invasive ventilation episodes during the same ICU admission and rounded to the nearest whole day. The team of physicians routinely collects CFS, SAPS-3, and SOFA scores.

Patients were then classified into two groups, according to whether they developed delirium during ICU stay. Delirium screening using the CAM-ICU (Confusion Assessment Method for the Intensive Care Unit [25]) was performed by our nursing staff every eight hours (once every shift), and, in case of doubt, it was discussed with the attending physician. This delirium assessment instrument is highly reliable in the hands of health care providers. It comprises four features assessing: acute change or fluctuation in mental status, inattention, disorganized thinking, or an altered level of consciousness. To be diagnosed with delirium, the patient needed to have a RASS (Richmond Agitation Sedation Scale, [26]) score above -3, a positive CAM-ICU (defined as an acute change or fluctuation in mental status, accompanied by inattention, and either disorganized thinking or an altered level of consciousness) [27–29].

Our standard of care implies applying a protocol for the prevention, diagnosis, and early treatment of delirium based on the ABCDEF bundle on all patients [30]. Thus, our first objective is to optimize prevention measures by assessing analgesia, sedation, and care, trying to maintain multimodal analgesia with the lowest possible opioid dose, maintaining the lowest possible sedation except where deep sedation is necessary, and promoting early mobilization, optimizing the environment, and promoting the family' presence.

## Statistical analysis

Discrete variables were expressed as a number and percentage, while continuous variables are expressed as medians (with interquartile range). Variables were explored to evaluate their normal distribution using the Kolmogorov-Smirnov test. Firstly, Pearson's or Chi-square test (alternatively Fisher's exact test for expected values<5) or Mann-Whitney's U were used to perform the exploratory analyses to find significant differences. Variables with a p-value under 0,05 were taken as statistically significant. Finally, odds ratios (OR) with 95% confidence interval (CI) were used when comparing characteristics between patients who had developed delirium and delirium-free patients during ICU stay.

Secondly, we performed a univariate analysis of delirium incidence. Thirdly, a multivariate logistic regression analysis was carried out to quantify the existing associations. The multivariate analysis included all variables that proved significant associations with a p-value under 0.10 in the univariate analysis, considering an alpha error of 5%. Lastly, we performed a recursive partitioning test employing a CHAID (Chi-square Automatic Interaction Detection) classification tree. Statistical analyses were performed using the SPSS software package version 20.0 for Windows.

## Results

During the study period, 1534 patients were admitted to our ICU. Seventy-two patients were excluded from the statistical analysis (due to loss of data related to hospital transfer), obtaining a cohort of 1462 adult patients. The demographic and clinical characteristics of the n = 1462 patients (already divided into two groups) are summarised in Table 1.

**Table 1. Demographics and clinical characteristics of the studied population.**

| | | Delirium | No delirium | p |
|---|---|---|---|---|
| Number of patients, n (%) | | 93 (6) | 1369 (94) | - |
| Age, yr, median (IQR) | | 71 (60–81) | 66 (55–74) | < 0,001 |
| Sex, n (%) | Male | 58 (62,3) | 790 (57,7) | 0,38 |
| | Female | 35 (37,6) | 579 (42,3) | |
| Clinical Frailty Scale, median (IQR) | | 3 (3–4) | 3 (2–3) | < 0,001 |
| Admission type, n (%) | Emergency surgery | 14 (15) | 196 (14,3) | 0,035 |
| | Scheduled surgery | 18 (19,4) | 437 (31,9) | |
| | Medical patient | 61 (65,5) | 736 (53,8) | |
| Main diagnosis on admission, n (%) | Acute respiratory failure | 22 (14) | 148 (10,8) | < 0,001 |
| | Postoperative | 25 (26,9) | 553 (40,4) | |
| | Sepsis | 19 (20,4) | 150 (11) | |
| | Acute coronary syndrome | 5 (5,4) | 202 (14,8) | |
| | Coma | 9 (9,7) | 42 (3,1) | |
| | Cardiac arrest | 4 (18,3) | 19 (1,4) | |
| | Other | 9 (9,7) | 254 (18,6) | |
| Comorbidities, n (%) | Cardiovascular | 59 (63,4) | 720 (52,3) | 0,04 |
| | Respiratory | 27 (29) | 328 (24) | 0,27 |
| | Renal | 50 (53,4) | 334 (24,4) | <0,001 |
| | Hepatic | 15 (16,1) | 220 (16,1) | 0,99 |
| | Cancer disease | 26 (28) | 489 (35,7) | 0,13 |
| | Endocrine | 45 (48,4) | 39 (2,8) | 0,086 |
| SAPS 3 score, median (IQR) | | 59 (49–66) | 45 (38–55) | <0,001 |
| SOFA score, median (IQR) | | 5 (2–8) | 1 (0–4) | <0,001 |
| Organ-supportive treatments | Invasive MV, n (%) | 58 (62,4) | 351 (25,6) | <0,001 |
| | Days under invasive MV, days (IQR) | 6,5 (3–15) | 2 (1–5) | <0,001 |
| | Reintubation, n (%) | 5 (5,4) | 13 (9,5) | <0,001 |
| | Non-invasive MV, n (%) | 10 (10,7) | 88 (6,4) | 0,11 |
| | CRRT, n (%) | 5 (5,4) | 64 (4,7) | 0,76 |
| Isolation | Preventive isolation on suspicion of MDR, n (%) | 27 (29) | 199 (14,4) | <0,001 |
| | Confirmed isolation due to positive MDR, n (%) | 14 (15) | 58 (4,2) | <0,001 |
| Need of prone position, n (%) | | 4 (4,3) | 8 (0,6) | 0,005 |
| Neuromuscular blockade, n % | | 6 (6,5) | 5 (0,4) | <0,00 |
| Organ failure, n (%) | Cardiovascular | 70 (75,3) | 424 (31) | <0,001 |
| | Respiratory | 63 (67,7) | 475 (34,7) | <0,001 |
| | Renal | 50 (53,8) | 334 (24,4) | <0,001 |
| | Hepatic | 13 (14) | 68 (5) | <0,001 |
| | Hematologic | 13 (14) | 102 (7,5) | 0,02 |
| Number of organ failure/s, median (IQR) | | 3 (2–4) | 0 (0–2) | <0,001 |

Yr = years; IQR = interquartile range; MV = mechanical ventilation; CRRT = continuous renal replacement therapy; MDR = multidrug-resistant bacteria.

Dashes indicate no data.

**Table 2. Short-term outcomes.**

|  | Delirium | No delirium | p |
|---|---|---|---|
| LOS ICU, days, median (IQR) | 7 (3–15) | 2 (1–4) | <0,001 |
| LOS hospital after ICU discharge, days, median (IQR) | 10 (5–18) | 6 (3–11) | < 0,001 |
| Unplanned readmission to ICU, n (%) | 7 (7%) | 40 (3%) | 0,014 |
| Mortality upon ICU discharge, n (%) | 0 (0%) | 47 (3,4) | 0,07 |
| Mortality upon hospital discharge, n (%) | 5 (5,4) | 39 (2,85) | 0,72 |

LOS = length of stay; IQR = interquartile range; ICU = intensive care unit.

Ninety-three patients developed delirium (incidence of 6.3%). Patients who developed delirium during ICU stay were older (p<0.001) and had a higher score in the CFS. In this group, reasons for ICU admission included pre-ICU emergency surgery or a medical admission (acute respiratory failure, sepsis, coma, or cardiac arrest). SAPS-3 (59 vs. 45, p<0.01) and SOFA score (5 vs. 1, p<0.001) on admission were higher in the delirium group. Patients with delirium also presented a higher incidence of cardiovascular (p 0.04) and renal comorbidities (p<0.001). Moreover, they required invasive mechanical ventilation in a higher percentage of the cases, plus they presented longer invasive mechanical ventilation duration (6.5 vs. two days, p<0.001). Regarding the development of organ failure(s), patients who developed delirium had a higher incidence of cardiovascular, respiratory, renal, hepatic, and hematological failures. Likewise, they required higher preventive (29% vs 14%, p<0.01) and confirmed (15% vs. 4%, p<0.01) isolation due to multidrug-resistant bacteria (MDR).

When studying outcomes (early prognosis, Table 2), patients who developed delirium presented longer ICU length-of-stay (7 vs. 2, p<0.001) and longer hospital stay once discharged from ICU (10 vs. 6, p<0.001). Furthermore, they presented an increased unplanned ICU readmission rate (7% vs. 3%, p<0.014). However, we did not find differences in mortality (upon ICU and ward discharge).

We initially performed a univariate analysis with all statistically significant variables: age above 74, Clinical Frailty Scale above 3, specific reason for admission (acute respiratory failure, sepsis, coma, cardiac arrest, urgent surgery), comorbidities (cardiovascular and renal), SAPS-3 score above 56, SOFA score on admission above 4, invasive mechanical ventilation, reintubation rate, preventive and confirmed isolation due to MDR, prone position, NMB, organ failures (respiratory, cardiovascular, renal, hepatic and hematological) and the number of failed organs above 2 (S1 Table).

In the multivariate analysis (Table 3), the use of neuromuscular blockade (OR 7.2, 95% CI 1.99–26.27) and the number of failed organs above 2 (OR 4.9, 95% CI 2.9–8.2) were the

**Table 3. Multivariate analysis.**

| VARIABLE | OR (95% IC) |
|---|---|
| Age above 74 yrs | 2,1 (1,3–3,5) |
| Coma on ICU admission | 2,5 (1,07–5,8) |
| > 2 organ failures | 4,9 (2,9–8,2) |
| Invasive MV | 1,9 (1,1–3,3) |
| Confirmed isolation due to MDR | 2,4 (1,2–4,6) |
| Neuromuscular blockade | 7,2 (2–26,3) |

Yrs = years; ICU = intensive care unit; MV = mechanical ventilation; MDR = multidrug-resistant bacteria.

strongest independent predictors of transitioning to delirium. We also observed high odds ratio for age above 74 (OR 2.1; 95% CI 1.3–4.5), coma as the reason for ICU admission (OR 2.5, 95% CI 1.1–5.8), need for invasive mechanical ventilation (OR 1.92, 95% CI 1.1–3.3) and confirmed isolation due to MDR (OR 2.4, 95% CI 1.2–4.8). Finally, we should clarify that most patients presented a toxic/metabolic/respiratory origin coma, and few were secondary to primary neurological problems.

Regarding the recursive partitions test using a CHAID classification tree, we observed a higher incidence of delirium in patients who presented more than two OF (20.4%) and patients with less than two OF, a higher incidence over 74 years of age (8.6%).

## Discussion

In this study, trying to find risk factors associated with delirium development, we found that the highest risk observed for developing delirium clustered in patients who presented more than two OF and patients over 74 years old. Furthermore, in our cohort, patients who developed delirium showed a longer ICU length of stay, a longer length of hospitalization after discharge from ICU, and an increased ICU readmission rate (7%), with no differences in mortality.

Considering the results we obtained in the multivariate analysis, patients in coma on admission to the ICU, patients who required invasive mechanical ventilation or continuous NMB, and patients who needed isolation due to an identified MDR were also at high risk of developing delirium. When compared with risk factors assessed in published studies [7–9], we highlight the lack of information regarding the relationship between the development of delirium and the use of NMB and between the development of delirium and contact isolation required after confirmation of the presence of MDR. The risk of delirium development observed in patients requiring NMB may be linked with the concurrent use of deep sedation. Still, one could raise a question: is one pharmacological group of NMBs more associated with delirium than another? On the other hand, the presence of MDR implies specific contact measures, making the family visit and contact with the patient more difficult, in addition to the need for broad-spectrum antibiotics, which could also play a role in the development of delirium.

Knowledge of the risk factors associated with delirium development among critical patients is essential for optimal patient management. In essence, it provides insight into a complex syndrome, facilitates the detection of high-risk patients, and allows us to improve prevention programs. One of the first attempts to identify those variables associated with an increased risk of delirium in critically ill patients [7] recognized the following with a strong level of evidence: trauma or emergency surgery before ICU admission, APACHE II score, coma, delirium on the previous day, use of mechanical ventilation and metabolic acidosis; multiorgan failure had moderate evidence (OR varying from 1.09 to 8.8). Regarding current guidelines [8], Devlin et al. found strong evidence for age, dementia or prior coma, pre-ICU emergency surgery or trauma, sex opioid use, mechanical ventilation, benzodiazepine use, and blood transfusion; while moderate evidence was found for a history of hypertension, admission due to neurologic disease, trauma and use of psychoactive medication. Whilst the APACHE score presented a strong association with delirium development, the SOFA score was inconclusive. Our study observed a statistically significant difference between SAPS-3 and SOFA scores when comparing non-delirium and delirium patients. When performing univariate analysis (S1 Table) we found an OR 4,07 (95% 2,65–6,23) for SAPS-3 score above 56 and an OR 4,05 (95% 2,64–6,21) for SOFA score above 4. Upon multivariate analysis, we observed an OR for multiorgan failure of 4.9, with a narrower confidence interval (95% CI 2.9–8.2) than the one observed by Zaal et al. [7].

The development of delirium is associated with a worse short-term prognosis, such as increased mortality, cognitive impairment, longer duration of mechanical ventilation, and longer length of stay in the ICU [5]. Even though we did not find differences in mortality, our patients who developed delirium presented more prolonged ICU and hospital length-of-stay (median of seven days in ICU, plus ten days in the wards). It should be noted that a longer duration in ICU is associated with an increase in morbidity and mortality, partially explained by potentially modifiable ICU factors such as the use of corticosteroids, neuromuscular blocking agents, benzodiazepines, or mechanical ventilation (already known risk factors for delirium) [31]. Along with this, and even though it was out of our study's scope, we must highlight that delirium is also associated with worse long-term prognosis, such as persistent cognitive impairment [32–34] and disability in activities of daily living, including worse motor-sensory function [35, 36].

We would also like to focus on the associated increase in ICU readmission rate (7% in our cohort), similar to the results published in other studies [37, 38]. Besides, previous studies have shown risk factors associated with unscheduled admissions, such as indices of pre-existing ill-health, previous prolonged ICU length of stay, administration of steroids, need for blood products, need for extrarenal clearance techniques, or primary diagnosis of respiratory, gastrointestinal, metabolic or renal pathology [39–41]. Readmission to the ICU has proven frequent and strongly related to poor outcomes [42]. However, measures to prevent them remain elusive, as only a small percentage of readmissions are reported to be preventable [43].

This specific group of patients shows a high risk of unscheduled hospital readmissions and an increased risk of developing post-ICU syndrome, already known to profoundly affect patients' perceived quality of life [44–46]. Prolonged exposure to risk factors from the ICU environment, including delirium development (with the consequent risk of developing secondary functional disability), makes this group of patients very vulnerable. Improving our understanding of risk factors amenable to intervention could improve our clinical management, plus develop post-ICU care programs. It has, therefore, important implications for research and public health policies.

Our study has several potential limitations. The main one is the relatively low incidence of delirium in our cohort (6.3%). Previous reports [47–49] from mixed ICU populations have demonstrated an incidence ranging between 30 to 80% observed in studies involving exclusively mechanical ventilated patients. Possible reasons for our low incidence could be (1) not including cardiac, thoracic, nor neuro-surgery due to the characteristics of our ICU, (2) an underdiagnosis of the condition (loss of cases related to undiagnosed hypoactive delirium [50]), (3) fluctuating handling of CAM-ICU screening tool likely related to frequent staff changes within the nursing pool or (4) optimized analgosedation management and preventive measures, which have been improved in recent years (although we do not have reliable data before 2016 to be able to compare whether this hypothesis is true). Delirium screening scales have the limitation of tagging but not necessarily identifying delirium. PADIS [8] supplemental material brings to light the controversy between delirium assessment reliability and sedation or consciousness, suggesting that the level of arousal could influence delirium assessment. CAM-ICU performed in routine practice has high specificity but low sensitivity, hampering early detection of delirium [51].

Two distinct clinical states (sedation and delirium, both associated with morbidity and mortality risks) can appear as a positive CAM-ICU screen and are considered equivalent to a 'delirium diagnosis.' However, sedation-associated positive delirium scores that normalize when sedation is lightened [52] confer no greater risk than documented in critically ill adults without delirium. Moreover, transient CAM-ICU positivity in the context of deep sedation behaves very differently than CAM-ICU positivity or ongoing delirium symptoms with RASS levels of, or near, 0.

Two other limitations to consider are the study's unicentric character and the somewhat limited number of patients, and the reliance on medical record data, introducing the potential for missing data. Regarding these two statements, in the first place, we believe the unicentric character could favor data homogeneity and consistency of the applied management. In the second place, although a lack of data is real, our ICU medical team tries to reduce this bias as much as possible by carrying out periodic reviews of the database.

The present study also has several strengths. We managed to include a high sample size considering our relatively small ICU capacity (between 8 and 10 available ICU beds), and we applied a reasonably new statistical model, which allowed us to detect the group most at risk of developing delirium. Different lines of action have been generated from the study, targeting a specific high-risk group and implying a change in our day-to-day work. Firstly, we have reinforced prevention measures (ABCDEF bundle) both within the ICU and within the hospital wards (thanks to our ICU outreach team) and encouraged a rigorous and systematic use of screening tools (CAM-ICU) within our ICU population. Furthermore, we have reinforced the presence of family members within the ICU, extending visiting hours, providing psychological screening and support, increasing the awareness of possible long-term consequences of intensive care among ICU survivors, and engaging them in the care of their relatives.

Secondly, and thanks to the great collaboration of our nursing team, high-risk patients are closely followed-up once discharged through the Continuity-of-Care Nursing team; this has led to our first multidisciplinary protocol for the management of post-ICU syndrome (coordinating both the hospital team and the Primary Care health centers attached to the hospital area to which we belong). During hospital admission, we support a nurse-led follow-up lead, coordinating with healthcare professionals and resource planning, and organizing a ward-discharge plan with the corresponding level of health care, guaranteeing the continuity of care. Our post-discharge follow-up program assures a satisfactory hand-off with the hospital ward team and discusses the next steps with the patient and family. They also provide a support program for families and caregivers, keeping in mind the patient's values and wishes in the shared decision-making process. As for the post-discharge recovery, the principal targets are to return the patient to baseline by promoting continuous care, sharing knowledge, professional experience, and resource availability among professionals at all levels of care. A comprehensive assessment of the patient at the Primary Care Provider is performed. An evaluation is carried out at intervals defined by the Primary Care and Continuity-of-Care Nursing team [53, 54]. It remains to be seen whether these changes affect long-term morbidity and mortality, including a decrease in unplanned hospital and ICU readmission rates.

## Conclusions

The detection of patients at high risk for developing delirium could imply a change in management and improved quality of care, emphasizing prevention measures, including close follow-up once discharged and collaboration with primary care. In our cohort, patients over 74 years old and those who presented more than two OF had the highest risk. Other identified relevant risk factors were coma on ICU admission, invasive mechanical ventilation, continuous NMB and patients who needed isolation due to identified MDR. Moreover, patients with delirium showed a more prolonged ICU and hospital length-of-stay and an increased CU readmission rate.

Our commitment towards critical care patients prompts us for early-diagnosis improvement through the systematic use of screening tools and the meticulous implementation of prevention programs. We must empower health professionals with information, education, and resources. The cornerstone would be achieving a multidisciplinary collaboration for managing these patients, improving their long-term prognosis and quality of life.

## Supporting information

**S1 Table. Univariate analysis.**
(DOCX)

**S1 File. Abbreviations list.**
(DOCX)

## Acknowledgments

We would like to thank all clinical members of the ICU team at H.U del Henares, who actively helped gather the data for the present article.

## Author Contributions

**Conceptualization:** Federico Gordo.

**Formal analysis:** David Varillas-Delgado.

**Investigation:** Beatriz Lobo-Valbuena, Sofía Garcia-Manzanedo, Maria-Mercedes Garcia-Arias.

**Methodology:** Federico Gordo.

**Project administration:** Federico Gordo, Rosario Molina.

**Resources:** Maria-Mercedes Garcia-Arias.

**Software:** David Varillas-Delgado.

**Supervision:** Federico Gordo, Rosario Molina.

**Validation:** Rosario Molina.

**Visualization:** Sofía Garcia-Manzanedo.

**Writing – original draft:** Beatriz Lobo-Valbuena.

**Writing – review & editing:** Federico Gordo, Ana Abella, Inés Torrejón.

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
