## [Decision Letter · Decision Letter 0]

8 Jun 2021

PONE-D-21-09224

Risk factors associated with the development of delirium. A prospective observational study

PLOS ONE

Dear Dr. LOBO VALBUENA,

Thank you for submitting your manuscript to PLOS ONE. After careful consideration, we feel that it has merit but does not fully meet PLOS ONE’s publication criteria as it currently stands. Therefore, we invite you to submit a revised version of the manuscript that addresses the points raised during the review process.

We look forward to receiving your revised manuscript.

Kind regards,

Aleksandar R. Zivkovic

Academic Editor

PLOS ONE

Journal Requirements:

2) Please update your title to reflect that the study assesses risk factors associated with delirium experienced in the ICU.

3)  Please provide additional details regarding participant consent. In the ethics statement in the Methods and online submission information, please ensure that you have specified:

 - whether consent was obtained

 - whether consent was informed

 - what type of consent you obtained (for instance, written or verbal, and if verbal, how it was documented and witnessed).

 - if your study included minors, state whether you obtained consent from parents or guardians.

 - if the need for consent was waived by the ethics committee, please include this information.

4) We note that you have included the phrase “data not shown” in your manuscript. Unfortunately, this does not meet our data sharing requirements. PLOS does not permit references to inaccessible data. We require that authors provide all relevant data within the paper, Supporting Information files, or in an acceptable, public repository. Please add a citation to support this phrase or upload the data that corresponds with these findings to a stable repository (such as Figshare or Dryad) and provide and URLs, DOIs, or accession numbers that may be used to access these data. Or, if the data are not a core part of the research being presented in your study, we ask that you remove the phrase that refers to these data.

Reviewers' comments:

Reviewer #1: Major and minor concerns:

- Current Guidelines describe a meta-analysis of existing risk factors from a great number of studies and cohorts. So, the topic isn’t new and a lot of studies on delirium risk factors are available. The reported risk factors in this study aren’t innovative and go in line with current guidelines. A systematic comparison with risk factors from current guideline could be an additional result to add value to the manuscript.

- A delirium incidence of 6.3% on an ICU is comparably low. Regarding a sensitive delirium screening (3 times a day with CAM-ICU), please describe your population more precisely, also the group of patients without delirium. Maybe your ICU population has lower risk for delirium compared to other populations (e.g. no heart surgery patients).

- Who performed Delirium screening and Clinical Frailty Scale? Are these routine data from clinical staff?

- Please put data from univariate analysis at least in a supplement.

- The discussion should focus more on the results from multivariate analysis. The conclusion that sicker and older patients have a higher risk for delirium is too little, because it is know from preexisting literature. What is new? What is the difference? What implementation strategies will be made to prevent delirium on their ICU?

It is a simple, well-executed prospective cohort description. Perhaps you can put some more work into comparing them to the existing literature and highlighting the specifics of their work.

Reviewer #2: Overall:

This is a somewhat clearly written manuscript outlining an observational study to identify risk factors of delirium. The project doesn’t really add anything new to the literature as no new risk factors have been identified. The low delirium rate in the population is problematic and the rationale for this has not been flushed out. Methodologic issues and limitations on reporting (standard of care, missing data) may have contributed to the findings. There are grammatical errors and word choices that impair understanding in some areas.

Abstract:

The abstract should clearly state that the dataset used is from a registry.

Tool used for delirium detection is not provided in the abstract.

Introduction:

The introduction is a bit disorganized and doesn’t support the need for the project well. The authors propose delirium is under-recognized in the ICU but do not provide a reference for this statement. The authors list a number of risk factors for delirium, but there isn’t a statement about whether these are adequate. The limitations on prevention and treatments for ICU delirium has been outlined, but these are not tied to the study and how additional risk factors could/would modify patient care paradigms and/or improve outcomes.

Methods:

Page 4, paragraph 1: Please define use of the term ‘polyvalent’. This doesn’t seem to be the correct word.

Page 4, paragraph 2: Please spell out the acronym RASS and provide a reference for this tool.

While the project study subjects enrolled prospectively, the reliance on medical record data introduces potential for a substantial amount of missing data and this has not been addressed.

A more detailed description of standard of care in relation to delirium should be provided. It would be helpful to know if the unit utilizes a ‘bundle’ (and which one) to prevent delirium which may contribute to the low delirium rate found in this study.

Did subjects receive a daily sedation break? This would be important for evaluating the project.

Please provide an operational definition of ‘organ failure’.

Potential collinearity between variables may be a problem with the analysis but it does not appear this has been evaluated.

Patients who are comatose can not be assessed with the CAM-ICU as they are not responsive. Subjects with coma on admission were included but it’s not clear how this was handled. It’s also not possible to assess patients for delirium while under complete neuromuscular blockade. How was the determination of delirium made or ruled out in these cases?

Results:

The delirium rate is very low for this cohort (6%). This should be discussed further within the manuscript.

Missing data, especially for delirium assessment/identification, should be reported.

Discussion and Limitations:

Discussion of the low delirium rate is not well developed. Further exploration of this is necessary.

Conclusions:

Conclusions are very brief and don’t add to the manuscript. It is primarily a restatement of results and call to action for health care providers.

Tables and Figures:

Tables are helpful and provide additional content complementing the text.

The figure is not viewable in the pdf. It is difficult to understand what is being presented or how it relates to the study.

Reviewer #3: Thank you for this interesting article dealing with risk factors for the delirium occurence in ICU patients. The article is straightforward written and of clinical importance. However, there are still some points that need to be addressed in order to improve the article. Also, I have some concerns to recommend the acceptance of the present article, since there is a large amount of delirious publications existing and it should be well explained, how this present article may add and supplement the understanding of the delirium etiopathogenesis. I would therefore suggest to majorily revise your manuscript.

- Headline should be more specific according to your study objectives. The setting should mentioned. Specify also the population on which your conclusion sould be drawn (general ICU patients, neurological/ surgical/ cardiovascular etc.).

- The term „APACHE II before admission“ should be more specific. (in which direction is delirium risk increased?)

- Correct „the use a bundle approach“, „its´“

- The second passage of the introduction should be better referenced after the second sentence.

- Abbreviations (e.g. SCCM, CAM-ICU, RASS) should be written out when first used. Please add a abbreviation list for specification (for e.g. in the supplementary materials).

- Inclusion and exclusion criteria sould be stated more profoundly. How was the willingness for study participation was ascertained when patients were sedated or could not communicate? Particularly in case of a delirium this is of major interest from an ethical point of view.

- The „new data protection regulation“ – what is meant by this term. It is enough to state that the study protocol was approved by the Ethics Committee (EC). Please add the number you received from the EC, accordingly.

- When did the CAM-ICU assessment take place. Please add a timeline/ timeframe. Who assessed the delirium state and how often was the assessment realized? What was the interrater-reliability like?

- How was the SAP3 score SOFA score assessed. Please integrate this in the method section and specify who assessed the scores (by experienced physicians?).

- p < .0001 should be changed to p < .001

- Please specify in the statistics sections: what is meant with the phrase „continuous variables were stratified…“. Please give an example. Also, „the cut-off point…“ was standardized. How was this standardized? What is meant by 0.1? Is it a p-value?

- Also the recursive partitioning test sounds to me a bit arbitrary. Could you please give a reference for this method. On which base where the variables for classification chosen? Based on the results of the multivariate analysis? What does the understanding of the CHAID classification add to the results?

- Please add „n = …“ when patient numbers are presented.

- „In this group, reasons for ICU admission included pre-ICU emergency surgery or a medical admission (acute respiratory failure, sepsis, coma, or cardiac arrest)“ – Where can the rate of pre-ICU emergency surgery be drawn from the table?

- Tables and figures should be presented chronologically after the manuscript text.

- Finally, it should be more clarified what is new and outstanding on this research topic. What change in delirium management can be conveyed from your results? What does this imply for future research?

- Give the reason why you state an evaluation of „moderate evidence“ for multiorgan function in the discussion section.

- How is the post-ICU follow-up realized? This sounds very progressive and sounds interesting for future research and to become routine clinical practice for prevention of long-term complications from ICU.

- Please speciy early aggressive treatment and other risk factors (other than what).

- How would you address the fact that delirium is associated with a higher risk for need of invasive mechanical ventilation? Vice verse, mechanical ventilation may increase the risk for delirium evolvement. How or in which direction would you evaluate the causative path, based on your data?

- Please let the Englisch language be checked via proof-reading by a native speaker.

- Please add line numbers on each page to facilitate the review process.

---

## [Author Response · Author response to Decision Letter 0]

5 Jul 2021

RESPONSE TO REVIEWERS

Reviewer #1

Current Guidelines describe a meta-analysis of existing risk factors from a great number of studies and cohorts. So, the topic isn’t new and a lot of studies on delirium risk factors are available. The reported risk factors in this study aren’t innovative and go in line with current guidelines. A systematic comparison with risk factors from current guideline could be an additional result to add value to the manuscript.

Thank you for your review, suggestions, and comments.

Following your recommendation, we have included a more extensive reference to the systematic review of risk factors carried out and published in the 2018 guidelines (with the statistical data provided in the guidelines as supplementary material).

A delirium incidence of 6.3% on an ICU is comparably low. Regarding a sensitive delirium screening (3 times a day with CAM-ICU), please describe your population more precisely, also the group of patients without delirium. Maybe your ICU population has lower risk for delirium compared to other populations (e.g., no heart surgery patients).

Following your advice, we have described in more detail the general characteristics of patients admitted to the ICU (as indicated, we do not have patients from cardiothoracic surgery or neurosurgery).

Who performed Delirium screening and Clinical Frailty Scale? Are these routine data from clinical staff?

Delirium screening is performed by our nursing staff, whilst CFS, SAPS-3 and SOFA scores are collected by the medical team. It’s part of our daily routine for several years now. Moreover, in case of doubt with any of the scores, we discuss it between the team members, and we collect in the Registry the agreed value.

Please put data from univariate analysis at least in a supplement.

We have included the data from the univariate analysis as Supplementary material – Supplementary Table 1. 

The discussion should focus more on the results from multivariate analysis. The conclusion that sicker and older patients have a higher risk for delirium is too little, because it is known from preexisting literature. What is new? What is the difference? What implementation strategies will be made to prevent delirium on their ICU?

Thank you for your comment. Following the suggestion, we have modified the discussion section by focusing more on the result of the multivariate analysis and possible strategies we could take to improve care for this group of patients.

 

Reviewer #2

The abstract should clearly state that the dataset used is from a registry. Tool used for delirium detection is not provided in the abstract. 

Thank you for your review and comments. We have modified the abstract, following your suggestion.

The introduction is a bit disorganized and doesn’t support the need for the project well. The authors propose delirium is under-recognized in the ICU but do not provide a reference for this statement. The authors list a number of risk factors for delirium, but there isn’t a statement about whether these are adequate. The limitations on prevention and treatments for ICU delirium has been outlined, but these are not tied to the study and how additional risk factors could/would modify patient care paradigms and/or improve outcomes.

We have modified the introduction and have also provided one reference (no. 6) supporting the statement of “underdiagnosed and underestimated problem”. 

Page 4, paragraph 1: Please define use of the term ‘polyvalent’. This doesn’t seem to be the correct word.

The term “polyvalent” has been replaced by “general”. 

Page 4, paragraph 2: Please spell out the acronym RASS and provide a reference for this tool.

We have followed your indications and included a reference (JAMA 2003;289(22):2983-91). 

While the project study subjects enrolled prospectively, the reliance on medical record data introduces potential for a substantial amount of missing data and this has not been addressed.

Thanks again. We have considered your comments and addressed this potential limitation in our discussion. 

A more detailed description of standard of care in relation to delirium should be provided. It would be helpful to know if the unit utilizes a ‘bundle’ (and which one) to prevent delirium which may contribute to the low delirium rate found in this study.

Thank you for your helpful comment. We have added a paragraph within “Material and Methods” briefly describing our protocol. 

Did subjects receive a daily sedation break? This would be important for evaluating the project.

Within our analgesia/sedation protocol we adjust sedation according to nursing needs (nursing-protocolized targeted sedation). We have adopted the term “dynamic sedation”, trying to keep patients with the least sedation possible, maintaining sedation levels around RASS 0 and -1 whenever possible.

Please provide an operational definition of ‘organ failure’.

We have included an operational definition of organ failure in “Material and Methods”, based on SOFA score above 2 for specific organs. 

Potential collinearity between variables may be a problem with the analysis but it does not appear this has been evaluated.

All the clinically relevant studied variables were included in the analysis. We did not intend to obtain a generalizable mathematical model.

Patients who are comatose cannot be assessed with the CAM-ICU as they are not responsive. Subjects with coma on admission were included but it’s not clear how this was handled. It’s also not possible to assess patients for delirium while under complete neuromuscular blockade. How was the determination of delirium made or ruled out in these cases?

During the administration of neuromuscular blockade, the CAM-ICU is not assessable given that deep sedation is maintained (RASS -5). CAM-ICU cannot be carried out until we are able to decrease sedation to a value of RASS above -3; patients who required deep sedation and NMB were able to fulfill CAM-ICU criteria at some point of their ICU admission. Regarding coma as main reason for admission, we should clarify that most of the comas were of toxic/metabolic/respiratory origin, and few were secondary to primary neurological status/pathology (given that we are not a reference center for neurocritical patients). We have, following your suggestion, added a small explanation regarding this last point. 

The delirium rate is very low for this cohort (6%). This should be discussed further within the manuscript.

We understand that one of the limitations of this study is, as you point out, the low incidence of delirium. Following your advice, we have added additional comments within the discussion.

Missing data, especially for delirium assessment/identification, should be reported.

We have followed your indications and addressed this potential limitation in our discussion. 

Discussion of the low delirium rate is not well developed. Further exploration of this is necessary.

We have broadened this section within the discussion.

Conclusions are very brief and don’t add to the manuscript. It is primarily a restatement of results and call to action for health care providers.

We have tried to complete the conclusions section following your feedback.

Tables are helpful and provide additional content complementing the text. The figure is not viewable in the pdf. It is difficult to understand what is being presented or how it relates to the study.

Considering the problems caused by the interpretation of the figure and given that the relevant information is reflected in the text, we have decided to remove the figure from the final version of the manuscript. We thank you for your input in this regard.

 

Reviewer #3

Headline should be more specific according to your study objectives. The setting should be mentioned. Specify also the population on which your conclusion should be drawn (general ICU patients, neurological/ surgical/ cardiovascular etc.).

Thank you for your review and comments. We have modified manuscript’s headline following your input. 

The term „APACHE II before admission“ should be more specific. (in which direction is delirium risk increased?)

APACHE (Acute Physiology and Chronic Health Evaluation) is a scoring system using routinely collected data and providing an accurate, objective description for a broad range of intensive care unit admissions, measuring severity of illness in critically ill patients. Previous studies regarding risk factors for delirium have found a strong association between high scoring in APACHE II and delirium development. We understand that in the introduction section, we mentioned apache II as a risk factor associated with the development of delirium. 

Correct „the use a bundle approach“, „its´“

We have made the correction.

The second passage of the introduction should be better referenced after the second sentence.

We have updated the references in that paragraph.

Abbreviations (e.g., SCCM, CAM-ICU, RASS) should be written out when first used. Please add a abbreviation list for specification (for e.g. in the supplementary materials).

We have corrected manuscript and provided a supplementary document with a list of abbreviations.

Inclusion and exclusion criteria should be stated more profoundly. How was the willingness for study participation was ascertained when patients were sedated or could not communicate? Particularly in case of a delirium this is of major interest from an ethical point of view. The „new data protection regulation“ – what is meant by this term. It is enough to state that the study protocol was approved by the Ethics Committee (EC). Please add the number you received from the EC, accordingly.

We thank you for your insightful comment. We have changed the paragraph on consent, providing more information related to the process, as well as inclusion and exclusion criteria. We have also provided the assigned number by the Research Ethics Committee of the university.

When did the CAM-ICU assessment take place. Please add a timeline/ timeframe. Who assessed the delirium state and how often was the assessment realized? What was the interrater-reliability like?

Following your advice, we have provided more information on how CAM-ICU was performed (by nursing staff, once every shift = once every eight hours). Moreover, our ICU protocol related to the early detection of delirium and optimization of preventive measures had been in place for several years prior to the study. At the time all ICU staff received appropriate training, in an attempt to standardize individual criteria. Also, in case of doubt, they were discussed with the attending physician.

How was the SAP3 score SOFA score assessed. Please integrate this in the method section and specify who assessed the scores (by experienced physicians?).

We thank you for your comment CFS, SAPS-3 and SOFA scores were routinely collected by the team of physicians. We have provided this information in the manuscript. 

p < .0001 should be changed to p < .001

We have modified it accordingly.

Please specify in the statistics sections: what is meant with the phrase „continuous variables were stratified…“. Please give an example. Also, „the cut-off point…“ was standardized. How was this standardized? What is meant by 0.1? Is it a p-value? Also the recursive partitioning test sounds to me a bit arbitrary. Could you please give a reference for this method. On which base where the variables for classification chosen? Based on the results of the multivariate analysis? What does the understanding of the CHAID classification add to the results?

Thank you for your feedback. We have tried to simplify the paragraph related to multivariate analysis. Recursive partitioning is a statistical method for multivariable analysis that creates a decision tree that strives to correctly classify members of the population by splitting it into sub-populations based on several dichotomous independent variables. CHAID is therefore a decision tree model that create classification systems that predict or classify future observations based on a set of decision rules. i.e., it involves a multivariate analysis where variables are automatically stratified according to importance, creating a risk map of different risk groups, allowing measures to be taken according to these groups.

Please add „n = …“ when patient numbers are presented.

Corrected. 

In this group, reasons for ICU admission included pre-ICU emergency surgery or a medical admission (acute respiratory failure, sepsis, coma, or cardiac arrest)“ – Where can the rate of pre-ICU emergency surgery be drawn from the table?

Regarding emergency surgery as admission type: 14.4% (n = 196) in the group of patients who did not develop delirium, and 15% (n = 14) in the group of patients who developed delirium during ICU admission. Data extracted from Table 1. 

Tables and figures should be presented chronologically after the manuscript text.

Following the journal's instructions for authors, we inserted the tables just after the paragraph that mentions them (“Tables should be included directly after the paragraph in which they are first cited”). 

Finally, it should be more clarified what is new and outstanding on this research topic. What change in delirium management can be conveyed from your results? What does this imply for future research?

We understand that delirium is a topic that has already been subject to many publications. Nevertheless, and in view of what has happened over the last year (with the exponential increase in the incidence of delirium among COVID-19 patients), we consider that it is still an unresolved and important issue, where more research is needed (both in prevention, early diagnosis, and treatment). Following your recommendation, we have broadened our discussion section. 

Give the reason why you state an evaluation of „moderate evidence“ for multiorgan function in the discussion section.

Moderate evidence regarding association of multiorgan failure and development of delirium was found in the study by Zaal et al (reference number 7). We simply record this statistical result in the manuscript, since we found a similar OR, with a narrower confidence interval.

How is the post-ICU follow-up realized? This sounds very progressive and sounds interesting for future research and to become routine clinical practice for prevention of long-term complications from ICU.

We would like to thank you for your comment, as it gives us the opportunity to explain our project. We have modified the last section of the discussion by adding a summary of our post-ICU patient care protocol, which was launched four years ago.

Please specify early aggressive treatment and other risk factors (other than what).

Following your indications, we have modified the paragraph and deleted the sentence, as it was misleading.

How would you address the fact that delirium is associated with a higher risk for need of invasive mechanical ventilation? Vice verse, mechanical ventilation may increase the risk for delirium evolvement. How or in which direction would you evaluate the causative path, based on your data?

Mechanical ventilation has been found to be an independent risk factor or delirium, although the specific pathogenesis is not well known. It may have to do with the effect of mechanical ventilation itself on intrathoracic and intravascular pressures, which ultimately exert a distant effect on the brain (probably with both physical and biochemical side effects). On the other hand, mechanical ventilation is associated with the need for sedation (both mild and deep), and this is also associated with an increased risk of developing delirium (especially if the drugs of choice are benzodiazepines or opioids).

Please let the English language be checked via proof-reading by a native speaker.

Following your comment, a native English speaker has revised the final version of the manuscript.

Please add line numbers on each page to facilitate the review process.

Thank you for your comment. We have inserted numbered lines in the manuscript.

---

## [Editor Report · Decision Letter 1]

19 Jul 2021

Risk factors associated with the development of delirium in general ICU patients. A prospective observational study.

PONE-D-21-09224R1

Dear Dr. LOBO VALBUENA,

We’re pleased to inform you that your manuscript has been judged scientifically suitable for publication and will be formally accepted for publication once it meets all outstanding technical requirements.

Kind regards,

Aleksandar R. Zivkovic

Academic Editor

PLOS ONE

---

## [Editor Report · Acceptance letter]

25 Aug 2021

PONE-D-21-09224R1 

Risk factors associated with the development of delirium in general ICU patients. A prospective observational study. 

Dear Dr. Lobo-Valbuena:

I'm pleased to inform you that your manuscript has been deemed suitable for publication in PLOS ONE. Congratulations! Your manuscript is now with our production department. 

Kind regards, 

on behalf of

Dr. Aleksandar R. Zivkovic 

Academic Editor

PLOS ONE